# Zinc Deficiency and Therapeutic Value of Zinc Supplementation in Pediatric Gastrointestinal Diseases

**DOI:** 10.3390/nu15194093

**Published:** 2023-09-22

**Authors:** Hsun-Chin Chao

**Affiliations:** 1Division of Pediatric Gastroenterology, Department of Pediatrics, Chang Gung Children’s Medical Center, Chang Gung Memorial Hospital, Taoyuan City 33305, Taiwan; hcchao1021@gmail.com; Tel.: +886-3-3281200; Fax: +886-3-3288957; 2College of Medicine, Chang Gung University, Taoyuan City 33302, Taiwan

**Keywords:** zinc, infectious diarrhea, IBD, celiac disease, peptic ulcer disease, GERD, children

## Abstract

The benefits of zinc in treating certain gastrointestinal (GI) diseases have been recognized for over two decades. This review aims to explore zinc deficiency (ZD) and the potential therapeutic value and safety of zinc supplementation in pediatric GI diseases. A systematic review of published articles on ZD and zinc as adjuvant treatments for GI diseases was conducted using various databases. Children with inflammatory bowel disease (IBD), celiac disease, and those receiving long-term proton pump inhibitor treatments are particularly susceptible to ZD. ZD in children with celiac disease and IBD is attributed to insufficient intake, reduced absorption, and increased intestinal loss as a result of the inflammatory process. Zinc plays a crucial role in maintaining the integrity of the gastric mucosa and exerts a gastroprotective action against gastric lesions. Although considerable evidence supports the use of zinc as adjuvant therapy for certain GI diseases in adults, its use is unspecified in children except for infectious diarrhea. Current evidence suggests that zinc supplementation with well-documented dosages helps reduce the duration of diarrhea in children with acute or persistent diarrhea, while there are no specific guidelines for zinc supplementation in children with IBD and celiac disease. Zinc supplementation appears to be beneficial in peptic ulcer disease or gastroesophageal reflux disease. The available evidence highlights the need for intervention programs to enhance zinc status and reduce the morbidity of certain GI diseases in children.

## 1. Introduction

For over two decades, researchers have explored the correlation between zinc and gastrointestinal (GI) diseases, as well as the potential benefits of zinc supplementation for specific GI conditions. In the last century, Semrad emphasized the crucial role of zinc in intestinal function, particularly in managing diarrhea [1]. Zinc plays a vital role in various cellular and systemic functions, including DNA repair, apoptosis, cell cycle progression, p53 activation, and the prevention of oxidative stress-induced DNA damage [2,3]. Zinc is considered a functional food for preserving GI mucosal function [4]. In terms of its epithelial barrier function, zinc could be used as an alternative to the use of steroids and anti-tumor necrosis factor modalities for treating inflammatory bowel disease (IBD) [5]. The intestinal mucosa not only experiences degeneration but also suffers severe effects from zinc deficiency (ZD). This results in a thinner mucus layer and changes in mucus composition, which were observed in both animal studies and human goblet cells [6,7].

ZD compromises GI epithelial function and is linked to various GI diseases, such as diarrhea, malabsorption, celiac disease, IBD, peptic ulcer disease (PUD), and gastroesophageal reflux disease (GERD) [8]. Zinc supplementation positively impacts many aspects of GI mucosa at both molecular and cellular levels, enhancing GI barrier function [9,10]. Numerous studies have shown that zinc supplementation improves epithelial barrier function, particularly via tight junction modifications [11,12]. Epithelial and endothelial tight junctions (zonula occludens) selectively seal the gap between adjacent cells, preventing unregulated paracellular exchange across the epithelial or endothelial barrier via zinc supplementation [13,14]. In addition to tight junction modification, epithelial barrier leak can be induced more significantly via induction of apoptosis and detachment [12].

This review aims to present evidence on the effects of ZD and zinc supplementation in pediatric GI diseases, with a focus on infectious diarrhea, GI inflammation, and GI mucosal damage. By examining the impact of zinc on antimicrobial effects, mucosal protection, and anti-inflammatory properties, this review aims to contribute valuable insights that can inform better therapeutic strategies involving zinc supplementation for specific GI diseases.

## 2. Materials and Methods

A literature search was conducted on PubMed, Medline, Cochrane, Embase, and Google Scholar for studies published between 1976 and 2023, using keywords related to zinc deficiency, zinc concentrations/levels, and zinc supplementation in specific GI diseases (infectious diarrhea, IBD focused on Crohn’s disease [CD] and ulcerative colitis [UC], celiac disease, PUD, and GERD). The specific objectives of this review were as follows: (1) to assess the risk of ZD and the impact of zinc treatment in different GI diseases; (2) to investigate the association between ZD and the effect of zinc supplementation in specific pediatric GI diseases; (3) to evaluate the safety and optimal dosage of zinc supplementation as adjuvant therapy for distinct classes of GI diseases; and (4) to explore any heterogeneity of response to zinc supplementation.

## 3. Results

### 3.1. Infectious Diarrhea

ZD can lead to intestinal hyperpermeability (leaky gut), resulting in increased nitric oxide and oxidative stress, which can cause diarrhea [15,16,17]. Studies in rats have shown that ZD upregulates the expression of intestinal uroguanylin, a peptide that stimulates chloride secretion driving fluid secretion, decreases the absorption of triglycerides by changing chylomicron development, and decreases the absorption of proteins by changing enterocyte peptidase activity, which can cause diarrhea [18,19,20,21].

In individuals with ZD, the GI tract may perhaps be one of the initially involved areas where symptoms manifest [22]. Such individuals are more susceptible to toxin-producing bacteria or enteroviral pathogens that activate guanylate and adenylate cyclases, stimulating chloride secretion and leading to diarrhea while also diminishing nutrient absorption. Additionally, ZD impairs the absorption of water and electrolytes, prolonging the duration of normally self-limiting GI disease episodes, and can exacerbate diarrhea caused by Vibrio cholera [23,24]. Notably, ZD can not only cause diarrhea but chronic diarrheal illnesses can also lead to ZD, thereby worsening diarrhea [25]. Studies have demonstrated that zinc supplementation influences lactulose excretion in children with stool isolates of *E. coli*, *Shigella* sp., and *Campylobacter jejuni*. The greatest reduction in lactulose excretion was observed in zinc-supplemented children who were lighter (weight-for-age less than 80%), thinner (weight-for-height less than 85%), and undernourished [middle upper arm circumference less than 12.5 cm], or with hypozincemia (less than 14 μmol/L) [26]. Zinc has also exhibited a direct antimicrobial effect on pathogenic enteric bacteria such as Salmonella and Shigella [27]. Children with acute diarrhea treated with zinc have shown a decrease in the duration and rate of diarrhea, as well as a reduced need for antibiotic therapy compared to controls [28].

A systematic review investigating the therapeutic value of zinc supplementation in acute and persistent diarrhea found evidence supporting the use of zinc to treat diarrhea in children; however, there was some uncertainty due to heterogeneity across the studies regarding zinc supplementation’s effect on diarrhea outcomes [29]. Zinc supplementation reduced the mean duration of diarrhea by 19.7% but had no significant effect on stool frequency or output while increasing the risk of vomiting [29]. Nevertheless, zinc supplementation has proven effective in decreasing the prevalence, morbidity, and mortality of diarrhea in healthy children in developing countries [29,30]. In cases of infectious diarrhea where there is typically increased intestinal permeability (lactulose/mannitol ratio), zinc supplementation has shown improvement [29,30,31]. Another systematic review of clinical trials involving 6165 children (aged >6 months) with diarrhea found that zinc supplementation reduced diarrhea duration and lessened diarrhea at days 3 and 7 in children with acute diarrhea, and also reduced the duration of persistent diarrhea. No serious adverse events were reported, but vomiting was more common in zinc-treated children with acute diarrhea [32]. A 1-year observational study of 20,246 children (aged 3–59 months) with diarrhea observed that regurgitation and/or vomiting occurred in 4392 (21.8%) cases. However, this phenomenon was transient and did not affect the continuation of zinc treatment [33]. According to meta-analyses of randomized, controlled intervention trials on children, the World Health Organization (WHO) and United Nations Children’s Fund (UNICEF) advocate for the regular use of zinc in treating children under the age of five with acute diarrhea, irrespective of its cause. They also advise diligent monitoring for any adverse events linked to zinc administration, with a specific focus on excessive or unusual vomiting.

A systematic review investigating the disease-specific and all-cause mortality attributable to ZD in children <5 years old found that ZD contributes to morbidity and mortality, especially in cases of diarrheal illness. The findings specify that zinc supplementation, provided as an adjunct treatment for diarrhea, may be the best way to target children most at risk of ZD [34]. Another systematic review, with an emphasis on developing countries, found that zinc supplementation could reduce the average duration of childhood diarrhea by approximately 20% [29]. A Cochrane database systematic review (2016) concluded that zinc may be beneficial for children aged six months or more in areas where the prevalence of ZD or malnutrition is high, while evidence does not support the use of zinc supplementation in children less than six months of age, in well-nourished children, and children at low risk of ZD [32].

In light of this robust evidence, zinc supplementation is considered a cost-effective method to reduce the duration of acute diarrhea in children [35]. Guidelines suggest a dosage of 20 mg/day for children older than 6 months or 10 mg/day for children younger than 6 months for at least 10–14 days during diarrhea [36,37,38,39].

### 3.2. IBD

IBD is a chronic condition characterized by inflammation in the gut mucosa and may appear at a young age, persisting throughout life with phases of recurrence and remission. This condition significantly impacts the quality of life of affected individuals [40,41]. Patients with IBD are at a high risk of experiencing nutritional deficiencies due to long-term gut inflammation and reduced oral intake. ZD is common among patients with IBD, with a prevalence rate ranging from 15 to 45% [42]. Given the involvement of immune function in IBD, maintaining normal zinc levels in these patients appears to be crucial.

Most reports of ZD in IBD come from Western countries and primarily concern adult patients. ZD is more frequently observed in patients with CD compared to those with UC. This difference is likely because zinc is predominantly absorbed from the GI lumen in the small intestine, particularly in the distal duodenum and proximal jejunum, which is affected in CD [40,41,42,43]. On the other hand, UC characteristically does not manifest in the small bowel. Several studies have shown that well-nourished UC patients did not experience ZD. When comparing patients with moderate or severe UC to healthy controls or patients in remission, their serum zinc concentrations were found to be normal or even elevated [44,45].

Elevated serum zinc concentrations in UC are correlated with increased levels of complement component C3C and elevated antinuclear antibodies during flares. These are known predictors of UC flares and markers of steroid dependence in UC [46,47]. ZD in UC patients is generally associated with malnutrition resulting from poor oral intake due to systemic illness during active UC, rather than being solely caused by the damaged colon [48,49]. During active UC flares, there is a correlation between increased serum zinc concentrations, increased levels of complement component C3C, and elevated antinuclear antibodies, which are recognized predictors of UC flares. The elevated serum zinc concentrations may potentially result from zinc release triggered by the activated inflammatory cascade. Nevertheless, one study did not find a correlation between serum zinc levels and severe inflammation (indicated by elevated erythrocyte sedimentation rate or C-reactive protein) in patients with UC [50]. In animal models of UC, ZD exacerbated the disease activity index, serum TNF-α levels, weight loss, and further shortening of the colon length. Several studies have explored the effects of zinc supplementation on inflammation in UC. For instance, Di Leo et al. reported improvements in diarrhea and weight gain in a rat colitis model when supplemented with zinc, although no effect was observed on neutrophil infiltration or visible inflammation [51]. On the other hand, Mulder et al. reported no change in the disease activity index or inflammation in human UC intestinal biopsies after zinc supplementation [52]. Zinc supplementation has the effect of suppressing colitis in a mouse model, as proved by lessened disease activity and histological severity, as well as decreased myeloperoxidase activity [53]. The effects of ZD on the development and progression of colitis were observed in mice; ZD can aggravate colonic inflammation via activation of the IL-23/Th17 axis [54].

ZD is a well-documented issue in individuals with CD due to a reduction in zinc absorption within the small intestine, as well as chronic dietary restrictions and intolerances [55,56]. In a substantial cohort of patients diagnosed with IBD, approximately 8.5% were estimated to have insufficient zinc intake, and as many as 29.3% exhibited deficient levels of serum zinc. Notably, ZD was even observed in individuals with CD during periods of remission [57]. ZD was even documented in CD patients during remission [58]. CD patients receiving total parenteral nutrition can develop acute ZD, resulting in acrodermatitis enteropathica and decreased vision [59]. More commonly, ZD in CD contributes to stunted growth in children and manifests as decreased taste sensation, visual acuity, and immune function [60,61] A positive correlation between increased mucosal permeability and CD disease activity was reported, as confirmed by the increased uptake of large molecular markers (such as lactulose) from the GI lumen into the bloodstream [62]. During periods of remission, heightened transmucosal permeability was employed as an indicator to predict the relapse of CD [63].

In a human intestinal cell line (CACO-2/T7), when zinc levels are reduced and these cells are exposed to TNFα (which is often present in the inflamed mucosa in CD), it leads to increased apoptosis and disrupts the organization of tight junctions, compromising the integrity of the epithelial barrier [64]. This helps explain why zinc supplementation is reported to decrease transmucosal leak in CD [65,66]. ZD may have an effect on the tight junction of intestinal epithelial cells, subsequently allowing more neutrophil migration and luminal antigen permeation [11]. Zinc supplementation may be beneficial in CD, whether the disease is active or in remission. It was postulated that zinc supplementation in patients with IBD would increase serum and mucosal levels of the zinc-dependent free radical scavengers superoxide dismutase [52]. Brignola et al. discovered that in patients with active CD, zinc supplementation significantly enhanced not only serum zinc levels but also the levels of zinc-dependent hormones such as thymulin. Thymulin has the potential to enhance T-cell differentiation and natural killer cell activity [66,67,68,69].

Two epidemiological reports validated that regular zinc intake via diets can reduce the risk of developing IBD [67,69], confirming zinc’s protective role against the condition. However, ZD was associated with adverse disease-specific outcomes, subsequent hospitalizations, and surgeries in IBD patients [68]. Despite these findings, the relationship between serum zinc levels and clinical activity in IBD patients remains inconclusive [69,70]. Clinical investigations on the efficacy of zinc supplementation for treating IBD are lacking, but earlier studies on animal and cell models have shown promising results, warranting further clinical studies to examine the efficacy of zinc treatment on clinical activity, inflammatory markers, and mucosal healing.

A high prevalence of micronutrient deficiencies, particularly zinc, was reported in pediatric IBD patients [71,72,73,74,75], with implications for poor clinical outcomes such as subsequent hospitalizations, disease-associated complications, and an increased risk of surgeries [74]. In a retrospective study that investigated the prevalence of ZD in children under 17 years with IBD, subjects were categorized into CD (n = 98), UC (n = 118), and normal controls (n = 43). The results revealed significantly lower serum zinc levels in CD (median, 64 μg/dL) compared to UC (median, 69 μg/dL) and normal controls (median, 77 μg/dL). Additionally, the prevalence rate of ZD was significantly higher in CD (60.2%) than in normal controls (37.2%), but not significantly different from UC (51.7%) [73]. Another study conducted on 165 pediatric patients with IBD (under 17 years; 87 CD and 78 UC) surveyed the prevalence and predictors of anemia and micronutrient deficiencies at diagnosis and 1-year follow-up. The prevalence of zinc deficiency was found to be 10% at diagnosis and 6% at follow-up, while anemia was prevalent in 57% of patients at diagnosis and 25% at follow-up [74].

In a larger study involving 359 children (median age 14.1 years) diagnosed with IBD (240 CD, 119 UC) and followed up for a median of 7 years, a higher prevalence of zinc deficiency was observed in patients with CD at diagnosis and last follow-up compared to UC patients. The prevalence of ZD in CD patients at diagnosis and last follow-up was 53% and 11.5%, while in UC patients, the prevalence was 31% and 10%, respectively [68]. A retrospective cohort study by Sikora SK et al. investigated serum levels of iron, zinc, folate, selenium, vitamin B12, vitamin A, and vitamin E in 154 pediatric patients (mean age 11.27 ± 3.74 years; 83 males) with inflammatory bowel disease (IBD) (74 with UC and 80 with CD). The study found that serum zinc levels were significantly lower in patients with CD (8.74 ± 2.08 µmol/L) compared to patients with UC (11.33 ± 4.16 µmol/L) and patients without IBD (11.49 ± 1.63 µmol/L) [75].

A prospective cross-sectional study assessed the serum levels of trace elements in 74 children with IBD (38 with UC and 36 with CD) and 40 age-matched controls. The results showed that children with IBD had abnormal serum levels of trace elements, which were more significant in those with CD. Children with CD had a significantly lower serum zinc level (11.01 ± 2.49 mmol/L) compared to the control group (13.6 ± 1.63 mmol/L). The study indicated that the deficiency of trace elements in children with IBD likely arises from a combination of factors, including insufficient dietary intake, diminished absorption, and heightened intestinal loss resulting from the inflammatory process-related impairment of absorption. The decreased capacity of zinc to scavenge free radicals, caused by its deficiency, could potentially exacerbate the ongoing inflammatory processes associated with IBD [76].

ZD may complicate malabsorptive states in many GI diseases [77]. In a study by Griffin et al. to evaluate the effect of CD on zinc metabolism in adolescents (15 with CD [ages 8–18 years] and 15 healthy matched controls), the measures included zinc absorption, endogenous zinc excretion (feces and urine), and zinc balance [78]. Subjects were provided elemental zinc (12 mg/day) for 2 weeks and then underwent a 6-day metabolic study. In CD subjects, zinc absorption and plasma zinc concentration were significantly reduced compared to controls, but there were no significant differences in fecal or urinary zinc excretion. The zinc balance was significantly lower in CD than in controls. A systematic review analyzed the research on micronutrient deficiency in children with IBD and provided evidence-based guidelines for nutritional surveillance in this population. The results showed that ZD occurred at a higher rate in patients with CD than in healthy controls. The authors recommend annual zinc surveillance in pediatric patients with IBD, regardless of disease activity or phenotype [79].

Several animal studies observed that zinc supplementation potentially affects the healing processes associated with IBD. The number of perfused tight-junction complexes was significantly elevated in rats with colitis compared to controls, but this effect was not observed in rats with colitis given a high dose of zinc. These findings suggest that zinc might play a role in regulating tight-junction permeability, potentially impacting the healing processes associated with IBD [80]. ZnO nanoparticles have dose-dependent effects on alleviating dextran sulfate sodium-induced ulcerative colitis in mice, involving antioxidant and anti-inflammatory actions, suggesting its potential as a medical adjunct for IBD therapy [81].

Although recent literature indicates the potential benefits of zinc supplementation for CD, zinc products have not undergone FDA inspection and validation specifically for CD. There are no specific guidelines regarding zinc supplementation for CD beyond emphasizing the significance of maintaining a balanced and adequate nutritional intake. The daily intake of zinc above the current recommended daily allowance is suggested in specific GI diseases at risk of ZD.

In summary, the evidence from epidemiological studies indicates that regular zinc intake via diets can reduce the risk of developing IBD. ZD poses significant risks for adverse disease-specific outcomes, hospitalizations, and surgeries in IBD patients. Addressing ZD in patients with IBD, particularly those with CD, seems to be crucial in managing the condition and maintaining overall health. Zinc supplementation may offer potential benefits in managing inflammation in UC, but further research is needed to fully understand its effects and mechanisms of action. The prevalence of ZD is notably high in pediatric IBD patients, underscoring the importance of addressing nutritional deficiencies in this vulnerable population.

### 3.3. Celiac Disease

Patients with celiac disease exhibit deficiencies in various vitamins and minerals, including zinc [82,83]. Research has shown that both untreated and clinically remitted celiac disease patients have reduced plasma zinc levels [84]. ZD was documented with a prevalence as high as 67% in newly diagnosed adult patients with celiac disease, and it can affect up to 64% of pediatric patients with the condition [85,86]. ZD was found to correlate with villous atrophy in celiac disease patients. A study reported that 60% of patients with partial villous atrophy, 80% with subtotal villous atrophy, and 92% with total villous atrophy were deficient in zinc [87]. Stenberg et al. proposed that ZD, particularly in the intestinal mucosa, may activate the enzyme transglutaminase-2 (TG2), which is normally inhibited by zinc. This activation, in turn, leads to the formation of a TG2-thioester intermediate–deamidated gliadin complex, which acts as a “neo-antigen”, triggering an immune response in genetically susceptible individuals, resulting in inflammation and villous atrophy [88].

The management of celiac disease primarily involves a lifelong gluten-free diet (GFD). However, numerous studies have reported nutritional deficiencies and imbalances associated with GFD. In pediatric patients, studies have shown decreased intakes of magnesium, zinc, selenium, and folate on a GFD, leading to potential nutritional inadequacies in this population [89]. Regardless of the GFD, children with celiac disease are at risk of insufficient calcium, vitamin D, iron, and fiber intake [90]. These imbalances may be exacerbated during GFD. Specifically, children with celiac disease on a GFD may experience altered intake of magnesium, zinc, folate, and high glycemic index foods.

A narrative review examining celiac disease patients on a long-term GFD with good compliance (over 2 years) revealed that micronutrient deficiencies were common, with up to 40% of subjects experiencing zinc deficiency [91]. Children with untreated celiac disease and enteropathy exhibited significantly reduced serum zinc levels, which subsequently returned to normal upon adopting GFD [92,93]. Supplementation of zinc along with GFD did not result in additional increases in plasma zinc levels [94]. While certain studies did not detect variations in zinc absorption between individuals with untreated celiac disease and control groups, there is evidence suggesting disrupted zinc homeostasis in celiac disease patients [95,96]. Improved zinc turnover and reduced loss of endogenous zinc were observed after starting a GFD. Additionally, the “exchangeable zinc pool” (zinc pools in the body that can exchange with serum zinc) was found to be significantly decreased in celiac disease patients [96,97]. Interestingly, some celiac disease patients may experience ZD despite having normal zinc absorption [95].

In a study aimed at determining serum zinc values in 140 samples from 78 children with celiac disease in different phases, abnormally low serum zinc levels were observed in children with acute celiac disease (50% below 2 SD), but not in children receiving a GFD [98]. The results of the study suggest regular measurement of serum zinc concentration in children with celiac disease and support zinc supplementation in patients with reduced zinc values during a period of 2–4 weeks, as ZD could inhibit the recovery of the intestinal mucosa.

Data regarding the effect of zinc supplementation with GFD on the normalization of serum or plasma zinc in celiac disease patients are still limited and controversial. The absence of advantages from zinc supplementation in terms of raising plasma zinc levels in patients adhering to a GFD could be associated with various factors, including the extent of mucosal healing following the initiation of GFD, the pharmacokinetics of zinc, its distribution and storage within the body, as well as the duration of supplementation. Theethira et al. suggested measuring the serum zinc levels of celiac disease patients at diagnosis and repeating the procedure regularly until the levels were normal; zinc supplementation (25–40 mg/day) was recommended until zinc levels were normalized [99].

Significantly lower plasma zinc concentrations were found in celiac children with chronic diarrhea compared to healthy children [100]. ZD can be used as an indicator to suggest the diagnosis of celiac disease in children with short stature, low plasma zinc levels were observed in 54.2% of cases due to celiac disease [101]. Hambidge et al. observed an association between poor growth and unsatisfactory zinc status assessed by low hair-zinc levels [25]. The same association was reported by Halsted et al. in Iranian boys [102]. ZD (decreased plasma zinc levels) was reported in 71.4% of pediatric series with celiac disease (n = 134). In that study, patients with ZD (n = 96) were randomly treated with GFD plus zinc (n = 48) and GFD only (n = 48). The results showed that the group with GFD plus zinc (20 mg elemental zinc) for 4 weeks provided no additional benefit regarding the rise in plasma zinc concentrations. Both groups showed a significant increase in plasma zinc levels after starting GFD for 4 weeks [103].

In a study to evaluate the effect of zinc supplementation on serum zinc levels in children (n = 30) with celiac disease, the serum zinc level was determined at inclusion and zinc supplementation was given for 3 months, with zinc levels measured at 3 and 6 months. The results showed that serum zinc levels were significantly lower in celiac disease patients (0.64 ± 0.34 mg/mL) than in controls (0.94 ± 0.14 mg/mL), whereas they were nonsignificant in the old cases. During the follow-up, the serum zinc level among severely malnourished and stunted cases was also significantly lower, irrespective of their treatment status [85]. A randomized controlled trial evaluating the effect of zinc supplementation on serum zinc, copper, and iron levels in patients with celiac disease observed that the increase in iron and zinc was significantly higher in the group with GFD plus zinc than in the group with GFD only [104].

In summary, the prevalence of ZD is notably high in celiac disease patients, ZD was found to correlate with villous atrophy in celiac disease patients. The beneficial effect of zinc supplementation as adjuvant to the GFD diet in Celiac disease is controversial, further research, including clinical trials investigating the efficacy of different dosages of zinc supplementation in treating celiac disease, is warranted to better understand the potential benefits of zinc in managing the disease.

### 3.4. PUD

Zinc is a crucial metabolic requirement for the growth and repair of squamous tissue [10]. Over the past three decades, researchers have observed zinc-mediated suppression of gastric acid production and improvement in gastric ulcer healing [105]. The therapeutic effects of zinc on PUD were described in both animal and human studies [106,107,108,109]. In animal studies, zinc supplementation did not affect the formation of gastric ulcers but was found to delay ulcer healing [107]. Human studies have shown that zinc offers protective action on the gastric mucosa [108,109]. In a clinicopathological study, a possible correlation between serum zinc levels and PUD was identified, along with a plausible mechanism for these findings [110]. The study observed lower serum zinc levels in patients with PUD compared to normal controls, but significantly increased zinc content in the gastric mucosa of PUD patients. These results indicate that the low serum zinc levels in peptic ulcer patients may be due to a positive shift of zinc from serum to the gastric mucosa.

The use of omeprazole, which inhibits gastric acid production and raises the pH in the gastroduodenal tract was documented to diminish zinc absorption in the small intestine in humans [111]. Joshaghani et al. reported a decline in serum zinc levels in males following 8 weeks of omeprazole usage, and a similar reduction in zinc absorption was observed when gastric acid production was inhibited by histamine-2 (H-2) blockers [112,113]. A study by Zhang et al. evaluated the role of Helicobacter pylori infection and serum zinc value in gastric diseases, finding that a similar rate of Helicobacter pylori infection was found in gastritis, peptic ulcer, and gastric cancer patients, while serum levels of zinc were significantly reduced in gastritis, peptic ulcer, and gastric cancer patients, compared with healthy controls [114]. The study illustrates that serum zinc level is an indicator of gastric protection against damage.

Frommer et al. demonstrated the effectiveness of zinc in treating gastric ulcers. However, other studies did not find a significant association between zinc supplementation and peptic ulcer treatment [115]. In one report, high-dose zinc supplementation (220 mg/day) showed no significant effect on peptic ulcers, but a better treatment effect was observed in patients with normal zinc levels [116]. Kirchhoff et al. reported that zinc administration (150 or 0.5 mg/kg/day ZnCl) could effectively raise luminal pH in rats, similar to proton pump inhibitor (PPI) medications, by eliminating secretagogue-induced gastric acid secretion without causing the side effects associated with PPIs, such as liver cytochrome P450 inhibition [117].

#### 3.4.1. Zinc-Containing Compounds and Their Gastroprotective Effect

Zinc was observed to generate a gastroprotective action against various tentative gastric lesions, suggesting its crucial role in maintaining gastric mucosal integrity [118,119]. Complexes containing zinc have shown a better ability to protect the gastric mucosa compared to individual compounds [116,119,120,121,122]. Table 1 summarizes the gastroprotective effect of various zinc-containing compounds in PUD.

##### Zinc Acexamate

A systematic review and meta-analysis of 13 randomized clinical trials (n = 757) evaluated the clinical efficacy of zinc acexamate in treating peptic ulcers, using placebo or H-2 receptor antagonist drugs as control groups [120]. The results indicate that zinc acexamate is an effective drug for peptic ulcer treatment. In a multicenter double-blind clinical trial involving 276 patients with rheumatic diseases and a history of peptic ulcer or intolerance to non-steroidal anti-inflammatory drugs (NSAIDs), participants were treated with one NSAID and one capsule (300 mg) of either zinc acexamate (n = 141) or placebo (n = 135) at a single nocturnal dose [121]. After a 4-week treatment period, 88% of patients treated with zinc acexamate and 66% with placebo exhibited completely normal endoscopy results (*p* < 0.0005). Zinc acexamate proved effective, reducing the incidence of gastric and duodenal ulcers by 92% compared to the placebo. The anti-inflammatory activity of zinc sulfate on gastric hazards was examined in rats with acute and chronic inflammation; combining zinc with other anti-inflammatory medications could offer synergistic benefits and minimize gastric risks, especially when used in conjunction with diclofenac [122].

##### Zinc Sulfate

In rats, a dose-dependent effect of zinc supplementation to protect duodenal lesions was observed. A 5-day pretreatment with zinc sulfate (50 or 150 mg/kg/day orally) significantly reduced the rate, number, and area of cystamine-induced duodenal lesions in rats. However, pretreatment with a dose of 15 mg/kg/day over a 5-day period or single administration of zinc sulfate (50 mg/kg) did not markedly reduce the above-mentioned lesion parameters [119].

Previous studies have demonstrated that a zinc sulfate salt can inhibit HCl generation at the cellular level of the parietal cell. A double-blind clinical trial involving 90 patients with gastric and duodenal ulcers investigated the effect of zinc sulfate on peptic ulcer treatment. A daily dose of 220 mg zinc sulfate was not significantly effective on peptic ulcer treatment, while patients with normal zinc levels showed better ulcer treatment outcomes [116].

Research investigated how two forms of zinc sulfate (monohydrate H_2_O and heptahydrate 7H_2_O) differ in their entry characteristics into parietal cells across various physiological conditions associated with acid secretion. Both monohydrate and heptahydrate zinc sulfate displayed a concentration-dependent cell entry, though heptahydrate zinc sulfate exhibited a significantly faster entry into parietal cells compared to monohydrate zinc sulfate. However, during resting conditions in fasted animals, monohydrate zinc sulfate displayed a quicker entry [123].

##### Zinc Monoglycerolate

The potential gastroprotective effects of zinc monoglycerolate (containing zinc >12 mg/kg), a slow-release zinc complex, were investigated in multiple rat models of gastric ulcers [124]. Zinc monoglycerolate demonstrated efficacy in preventing ulcer formation. An animal study comparing the effects of a ranitidine–zinc complex, and ranitidine alone utilized three experimental rat models (pyloric ligation, ethanol, and indomethacin) to assess their impact on gastric ulceration [125]. The results indicate that the addition of zinc does not disrupt the antisecretory effects of ranitidine on the gastric mucosa; instead, it might provide an extra layer of cytoprotection.

##### Zinc Gluconate

The role of zinc as an antiulcerogenic agent was ascertained in rats [106]. Pre-treatment with zinc gluconate at 10 mg/kg orally for three consecutive days could protect against alcohol-induced gastric epithelial damage and significantly prevent NSAID-induced gastric ulcers in rats. Zinc treatment increased the gastric mucosal barrier, enhancing the levels of mucus and hexosamine while decreasing acid output in gastric secretion. The results indicate that zinc appears to play a multifaceted protective role in chemically induced gastric ulcers.

##### Polaprezinc

Polaprezinc is a chelated form of zinc and L-carnosine, clinically used to treat gastric ulcers. It was determined that polaprezinc may be effective for treating human PUD [106,107,108,109,110,111,112,113,114,115,116,117,118,119,120,121,122,123,124,125,126,127]. Effects of polaprezinc on the mucosal ulcerogenic response induced by ammonia and monochloramine were examined in rat stomachs [127]. The results indicate that monochloramine damages the gastric mucosa at much lower concentrations than ammonia, and polaprezinc protects the stomach against injury caused by either ammonia or monochloramine. Polaprezinc has the potential to positively impact the process of ulcer healing via several mechanisms in rat models, including the mediation of zinc(2+) to enhance gastric microcirculation, exhibit antisecretory activity, and stimulate gastrin release. These effects may contribute to increased cell proliferation and differentiation during ulcer healing, ultimately leading to a trophic effect on the damaged gastric mucosa [128].

A study compared the efficacy of submucosal dissection-induced ulcer healing between PPI plus polaprezinc and PPI plus rebamipide treatments in adult patients, finding that PPI plus polaprezinc treatment showed non-inferiority to PPI plus rebamipide treatment in the ulcer healing rate at 4 weeks after submucosal dissection [129]. An animal study investigated the potential of polaprezinc for treating suspected gastric ulcers in dogs and assessed its impact on reducing acid-induced injury in canine gastric mucosa. The findings indicated that the introduction of HCl led to a time-dependent increase in gastric permeability and evident induction of apoptosis, as confirmed by immunofluorescence analysis [130]. Notably, the study revealed that polaprezinc did not significantly influence the rise in gastric permeability and did not provide protection against the development of apoptosis.

##### Zinc Carnosine

A study investigated 92 patients (67.4% males) with dyspepsia symptoms and a positive ^13^C-urea breath test, randomly assigning them to two groups: one group with a 14-day standard protocol of esomeprazole, amoxicillin, and clarithromycin, and the other group with a 10-day course of modified bismuth quadruple therapy fortified with zinc carnosine [131]. Results showed that 10 days of modified bismuth quadruple therapy fortified with zinc carnosine is superior to a 14-day standard protocol in eradicating H. pylori infection, with no additional significant adverse events.

##### Taurine Zinc

An animal study investigated the effect of taurine zinc solid dispersions (SDs) in ethanol-induced ulcers in rats, observing that taurine zinc (100, 200 mg/kg) SDs protected gastric mucosa from ethanol-induced injury. The gastroprotective effect was accompanied by a decrease in serum nitrite oxide and a significant increase in gastric prostaglandin E2 [132] In that study, taurine zinc at a dose of 200 mg/kg was found to provide more protection against ulceration compared to the same dose of taurine alone, implying a synergistic effect between taurine and zinc. These findings suggest that taurine zinc protects the gastric mucosa from ethanol-induced damage by enhancing antioxidant levels, reducing lipid peroxidation, and suppressing nitric oxide production.

In summary, zinc demonstrated its gastroprotective and anti-inflammatory properties in various experimental models, making it a promising treatment option for peptic ulcers. Zinc treatment increases the gastric mucosal barrier, enhancing the levels of mucus and decreasing acid output in gastric secretion. However, a dose-dependent effect to protect gastric or duodenal epithelial damage was observed in animal models. Further research is needed to optimize dosage and treatment regimens for improved clinical outcomes.

### 3.5. GERD

Excessive exposure to gastric acid is the major cause of GERD and reflux esophagitis. However, an increasing number of patients are experiencing insensitivity to PPI therapy, leading to a recurrence of symptoms. Consequently, finding alternative treatment options has become essential. It was observed that PPI use significantly reduces supplemental zinc uptake, resulting in decreased zinc body stores in males [117,133]. A study evaluating the effects of omeprazole treatment on trace element levels in GERD patients found that serum zinc levels were significantly lower in male patients after 8 weeks of omeprazole treatment, whereas no significant difference was observed in female patients [112]. As a result, it is suggested that zinc supplementation may be considered for male patients undergoing PPI treatment. Long-term PPI therapy in certain individuals, such as infants being treated for colic, may lead to reduced systemic levels of trace elements necessary for development, regeneration, and immune function [133].

Studies have shown that zinc supplementation can inhibit gastric acid secretion in both human and animal models. The authors of one study reported that exposure to a single dose of zinc salt raised intragastric pH for over 3 h, indicating that zinc supplementation effectively reduced secretagogue-induced gastric acid secretion and elevated luminal pH as effectively as PPI, without the side effects associated with hepatic cytochrome P450 inhibition caused by PPI [117]. A clinical trial involving oral zinc gluconate (containing 26.2 mg zinc, twice daily) for 2 weeks in GERD patients on long-term PPI therapy (>6 months) and healthy controls (not on any antacids or neutralizing medication) found that plasma zinc levels in healthy controls increased by 126% after zinc supplementation, compared to a 37% increase in those on long-term PPI therapy. The study also revealed that those with PPI therapy had a 28% lower plasma zinc level than healthy controls on their normal diet (without zinc supplementation) [133].

However, it is important to note that not all studies have shown positive results for zinc supplementation in GERD treatment [134]. A randomized double-blind study found that zinc supplementation could not significantly lessen the severity of GERD. The study included 140 patients (81 women, mean age 42.8 ± 11.5 years) divided into two groups: non-erosive reflux disorder and erosive reflux disorder. Each group was further divided into drug subgroups (treated with PPI, lifestyle changes, and 220 mg zinc daily) and placebo subgroups (treated with PPI, lifestyle changes, and placebo). Both drug and placebo groups showed a significant decrease in Reflux Disease Questionnaire (RDQ) scores after 3 months (*p* < 0.001), but the difference in RDQ scores between the drug and placebo groups was not statistically significant (*p* = 0.086) [134].

In summary, while some evidence suggests that zinc supplementation may be beneficial for male GERD patients undergoing PPI treatment, further research is required to fully understand its potential as an alternative therapy for GERD and reflux esophagitis. It is essential to consider individual patient factors and conduct well-designed clinical trials to determine the efficacy and safety of zinc supplementation in the treatment of GERD.

## 4. Zinc Supplementation: Dosage and Safety

The recommended daily intake (average requirement) ranged range from 6.2 to 10.2 mg/day for women with a reference body weight of 58.5 kg and from 7.5 to 12.7 mg/day for men with a reference body weight of 68.1 kg; the estimated average requirement ranged from 2.4 mg/day in infants aged 7–11 months to 11.8 mg/day in adolescent boys [135]. Recommended nutrient intakes for dietary zinc (mg/day) to meet the normative storage requirements from diets differing in zinc bioavailability ranged from 1.1 mg (high availability) up to 6.6 mg (low bioavailability) in infants aged 0–6 months and ranged from 3.3 mg (high availability) up to 11.2 mg (low bioavailability) in children aged 7–9 years months [136]. In adolescents, the average requirement ranged from 4.3 mg (high availability) up to 14.4 mg (low bioavailability) [136]. The upper limit level of zinc intake for an adult man is set at 45 mg/day and extrapolated to other groups in relation to basal metabolic rate. For children, the upper limit of intake is 23–28 mg/day [136].

Studies have demonstrated that higher doses of zinc can be considered safe due to its relatively short form in the body. However, it is essential to be cautious as excessive zinc intake can interfere with the absorption of iron and copper, potentially exacerbating deficiencies in these essential minerals. Therefore, the upper limit for daily zinc supplementation is set at 40 mg/day for adults.

In specific medical conditions such as IBD, higher doses of zinc have been recommended. For instance, in IBD patients in remission, a dosage of 40 mg/day for 10 days up to 110 mg three times a day for 8 weeks has been suggested [65]. Similarly, for individuals with celiac disease on a long-term GFD with good compliance, a recommended dosage of 25–40 mg/day of zinc has been proposed [91].

Research has shown that a total dose of zinc at 52 mg/day for 14 days had no adverse effects in patients with Barrett’s esophagus. On the contrary, it resulted in positive changes in intracellular signal transduction in Barrett’s epithelia, indicating potential benefits [137]. In the context of peptic ulcer disease, daily administration of 220 mg of zinc sulfate (equivalent to 50 mg elemental zinc) has been found to reduce ulcer size [112], in addition, those patients with normal levels of serum zinc had better healing processes.

However, caution must be exercised as prolonged intake of 150 mg/day of zinc can lead to toxicity, manifesting as anemia and severe copper deficiency [138,139]. Therefore, it is crucial to monitor serum copper levels regularly if patients are prescribed high daily doses of zinc (>50 mg/day) [140].

## 5. Heterogeneity of Response to Zinc Supplementation

Current evidence suggests that zinc supplementation can help reduce the duration of diarrhea in children with acute or persistent diarrhea. However, there is uncertain heterogeneity across studies regarding the effect of zinc supplementation on diarrhea outcomes. As of now, there are no specific guidelines for zinc supplementation in children with conditions such as IBD, celiac disease, peptic ulcer disease, or GERD.

Zinc supplementation appears to show potential benefits in managing conditions such as IBD, GERD, or peptic ulcer disease, but there is limited well-documented information on the appropriate dosage for pediatric patients. In the case of celiac disease, ZD has been found to correlate with the severity of villous atrophy. However, the effectiveness of zinc supplementation as an adjuvant to GFD remains controversial. While some experts have suggested zinc supplementation at a dosage of 25–40 mg/day in celiac disease patients until zinc levels return to normal, several pediatric randomized trials have indicated that zinc levels improve with GFD alone, irrespective of additional zinc supplementation [105,106,107].

## 6. Conclusions

Accumulating evidence has demonstrated that zinc supplementation has beneficial effects on various GI diseases. Numerous studies have also highlighted the association of ZD with the development of GI diseases, particularly via its negative impact on the epithelial barrier function of the GI tracts. ZD can lead to catastrophic consequences and exacerbate disease states by inducing barrier leakage. This review emphasizes that ZD is prevalent in pediatric IBD, with limited improvement during follow-up, and is more common in CD compared to UC. Adolescents with CD exhibit significantly reduced zinc absorption, worse zinc balance, and lower plasma zinc concentrations compared to healthy children.

Available evidence supports that zinc supplementation, within well-documented dosage limits, is generally safe and could serve as a prophylactic measure against various GI diseases, thus reducing morbidity in certain established conditions. Except for well-documented dosages for treating acute diarrhea, the treatment outcomes of using zinc supplementation as adjunctive therapy for specific GI diseases in children remain unclear. Numerous animal studies support the gastroprotective effect of zinc supplementation on gastric or duodenal mucosa in peptic ulcer models; further human studies are expected to accumulate more evidence. Additionally, investigating the optimal serum level of zinc and the most effective form of delivering zinc to distinct target tissues is essential to understand the precise mechanisms involved in its regulation of tight junctions.

## Figures and Tables

**Table 1 nutrients-15-04093-t001:** Gastroprotective effect of various oral zinc-containing compounds on peptic ulcer disease.

Zinc-Containing Compounds	Reference	Model	Advantages/Disadvantages
Zinc acexamate	[120,121,122]	Human/Rats	1. A 4-week zinc acexamate treatment (300 mg/day) proved effective in reducing peptic ulcer incidence compared to placebo (human) [121]. 2. Combining zinc acexamate with anti-inflammatory drugs provided beneficial additive effects and reduced gastric hazards (rats) [122].
Zinc sulfate	[116,119,123]	Human/animal	1. A 5-day pre-treatment with zinc sulfate (50 or 150 mg/kg/day) reduced the rate, number, and area of cystamine-induced duodenal lesions (rats) [116]. 2. A dose of zinc sulfate (220 mg/day) did not have significant effects on peptic ulcer treatment, while it had better ulcer treatment outcomes in patients with normal zinc levels (human) [119]. 3. Heptahydrate zinc sulfate exhibited a significantly faster entry into parietal cells compared to monohydrate zinc sulfate (rats) [123].
Zinc monoglycerolate	[124,125]	Animal	1. Dose-dependent zinc monoglycerolate (zinc > 12 mg/kg) in preventing peptic ulcer formation (rats) [124] 2. A similar protective effect against gastric damage was observed between the ranitidine–zinc complex (100 and 150 mg/kg) and equal doses of ranitidine (35, 70, and 105 mg/kg) in rats [125].
Zinc gluconate	[106]	Animal	A 3-day pre-treatment with zinc gluconate (10 mg/kg) protected against alcohol-induced gastric damage and prevented NSAID-induced gastric ulcers (rats) [106].
Polaprezinc	[126,127,128,129,130]	Human/animal	1. Treatment with polaprezinc (65 mg/kg/day) significantly raised the gastric luminal and mucosal levels of Zn2+ and significantly accelerated ulcer healing at day 7 upon ulcer induction (rats) [126]. 2. A randomized controlled study in adult patients (n = 210) with endoscopic submucosal dissection-induced ulcers were allocated to treatment with polaprezinc (150 mg/day) plus pantoprazole (40 mg/day) or treatment with rebamipide (300 mg/day) plus pantoprazole (40 mg/day). polaprezinc plus PPI treatment showed noninferiority to rebamipide plus PPI treatment in the ulcer healing rate at 4 weeks after submucosal dissection [127]. 3. Polaprezinc had the potential for treating gastric ulcers (dogs) [128].
Zinc carnosine	[131]	Human	Ninety-two adult patients with dyspepsia and positive ^13^C-urea breath test were randomly assigned into two groups. Ten days of modified bismuth quadruple therapy fortified with zinc carnosine is superior to 14 days of conventional treatment in eradicating H. pylori infection, with no additional significant adverse events.
Taurine zinc	[132]	Animal	Taurine zinc (200 mg/kg) protected against gastric ulcers more significantly than taurine alone, suggesting a synergistic effect between taurine and zinc (rats).

## Data Availability

Not applicable.

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
