# Peer review of "Zinc Deficiency and Therapeutic Value of Zinc Supplementation in Pediatric Gastrointestinal Diseases"

_nutrients, 2023, doi:10.3390/nu15194093_

Round 1
Reviewer 1 Report
The manuscript “Zinc Deficiency and Therapeutic Value of Zinc Supplementation in Pediatric Gastrointestinal Diseases” provides a comprehensive review of the literature on the occurrence of zinc deficiency during pediatric gastrointestinal diseases, which is particularly critical in inflammatory bowel diseases (IBDs) and discusses the beneficial effect of zinc supplementation for several gastrointestinal disorders. The topic is of great importance, as zinc deficiency often occurs during GIT diseases and is associated with the severity of the disease. Especially for IBD, which affects about 6.9 million people worldwide and is a growing health problem, concomitant therapy with Zn administration has been discussed for a long time, but the study situation is still insufficient, as the review summarizes well.
I believe this review will be of great interest to Nutrients readers and strongly recommend accepting after minor changes.
Lines 27-53: The introduction summarizes the effects of zinc deficiency on intestinal homeostasis very well. However, one important point is missing, as not only is the intestinal mucosa degenerated, but the mucus layer is also severely affected by zinc deprivation, leading to reduced thickness of the mucus layer and altered mucus composition, as observed in animal studies and human goblet cells (Quarterman, J., et al. (1976). "The effect of zinc deficiency on sheep intestinal mucin." Life Sci 19(7): 979-986; Quarterman, J., et al. (1973). "The Effect of Zinc Deficiency on Intestinal and Salivary Mucins." Biochemical Society Transactions 1(1): 101;Maares, M., et al. (2020). "Zinc Deficiency Disturbs Mucin Expression, O-Glycosylation and Secretion by Intestinal Goblet Cells." International Journal of Molecular Sciences 21(17): 6149). Even if "mucus" was not a key word in the literature search, the importance of mucus for gut health and especially the development of GIT diseases, such as IBD, is essential, and the review also discusses the effect of zinc supplementation on mucus (Lines 430-431). I therefore suggest including the effect of zinc deficiency on mucus and the respective studies at least in the introduction.
Lines 128-262: This section very nicely summarizes the current knowledge on the impact of zinc deficiency and supplementation during IBD. To mechanistically explain the impact of zinc on disease development, studies from Higashimura et al. and Li et al should be included (Higashimura, Y., et al. (2020). "Zinc Deficiency Activates the IL-23/Th17 Axis to Aggravate Experimental Colitis in Mice." J Crohns Colitis 14(6): 856-866; Li, J., et al. (2017). "ZnO nanoparticles act as supportive therapy in DSS-induced ulcerative colitis in mice by maintaining gut homeostasis and activating Nrf2 signaling." Scientific Reports 7(1): 43126.). For the sake of completeness, as this is a review of current studies on the impact of zinc for disease outcome, the following publications should also be mentioned: Ohkawara, T., et al. (2005). "Polaprezinc (N-(3-aminopropionyl)-L-histidinato zinc) ameliorates dextran sulfate sodium-induced colitis in mice." Scand J Gastroenterol 40(11): 1321-1327; Sturniolo, G. C., et al. (2002). "Effect of zinc supplementation on intestinal permeability in experimental colitis." J Lab Clin Med 139(5): 311-315.
Lines 494-495: Please specify if the reported study is based on an animal study or human data. It would also be interesting to know at this point how much zinc was administered.
Line 498-499: Is it 26.2 mg zinc or 26.2 mg zinc gluconate?
Line 524: I suggest including a newer reference for the recommended daily intake of zinc, as this reference is already outdated. Both WHO and EFSA have changed their recommendations depending not only on the gender but also on the zinc bioavailability of the respective diet consumed (i.e. depending on the phytate content) (EFSA Panel on Dietetic Products, Nutrition and Allergies (NDA). Scientific opinion on dietary reference values for zinc. EFSA Journal 2014 2014, 12; World Health Organization / Food and Agricultural Organization. Vitamin and Mineral Requirements in Human Nutrition. 2 ed.; World Health Organization: Geneva, Switzerland, 2004).
Lines 528-529: Please include a reference for the upper limit for zinc.
Line 85: I believe mumol/L should be µmol/L.
Line 113: Please change systemic to systematic.
Author Response
Reply to reviewer’s queries
Reviewer 1
The manuscript “Zinc Deficiency and Therapeutic Value of Zinc Supplementation in Pediatric Gastrointestinal Diseases” provides a comprehensive review of the literature on the occurrence of zinc deficiency during pediatric gastrointestinal diseases, which is particularly critical in inflammatory bowel diseases (IBDs) and discusses the beneficial effect of zinc supplementation for several gastrointestinal disorders. The topic is of great importance, as zinc deficiency often occurs during GIT diseases and is associated with the severity of the disease. Especially for IBD, which affects about 6.9 million people worldwide and is a growing health problem, concomitant therapy with Zn administration has been discussed for a long time, but the study situation is still insufficient, as the review summarizes well.
I believe this review will be of great interest to Nutrients readers and strongly recommend accepting after minor changes.
Answer: Thank you for your positive feedback and comments.
Q1. Lines 27-53: The introduction summarizes the effects of zinc deficiency on intestinal homeostasis very well. However, one important point is missing, as not only is the intestinal mucosa degenerated, but the mucus layer is also severely affected by zinc deprivation, leading to reduced thickness of the mucus layer and altered mucus composition, as observed in animal studies and human goblet cells (Quarterman, J., et al. (1976). "The effect of zinc deficiency on sheep intestinal mucin." Life Sci 19(7): 979-986; Quarterman, J., et al. (1973). "The Effect of Zinc Deficiency on Intestinal and Salivary Mucins." Biochemical Society Transactions 1(1): 101;Maares, M., et al. (2020). "Zinc Deficiency Disturbs Mucin Expression, O-Glycosylation and Secretion by Intestinal Goblet Cells." International Journal of Molecular Sciences 21(17): 6149). Even if "mucus" was not a key word in the literature search, the importance of mucus for gut health and especially the development of GIT diseases, such as IBD, is essential, and the review also discusses the effect of zinc supplementation on mucus (Lines 430-431). I therefore suggest including the effect of zinc deficiency on mucus and the respective studies at least in the introduction.
Answer: Thank you for your comments and valuable suggestions. I added the information about the effect of zinc deficiency on mucus in the introduction section (line 38-41, page 1).
Q2. Lines 128-262: This section very nicely summarizes the current knowledge on the impact of zinc deficiency and supplementation during IBD. To mechanistically explain the impact of zinc on disease development, studies from Higashimura et al. and Li et al should be included (Higashimura, Y., et al. (2020). "Zinc Deficiency Activates the IL-23/Th17 Axis to Aggravate Experimental Colitis in Mice." J Crohns Colitis 14(6): 856-866; Li, J., et al. (2017). "ZnO nanoparticles act as supportive therapy in DSS-induced ulcerative colitis in mice by maintaining gut homeostasis and activating Nrf2 signaling." Scientific Reports 7(1): 43126.). For the sake of completeness, as this is a review of current studies on the impact of zinc for disease outcome, the following publications should also be mentioned: Ohkawara, T., et al. (2005). "Polaprezinc (N-(3-aminopropionyl)-L-histidinato zinc) ameliorates dextran sulfate sodium-induced colitis in mice." Scand J Gastroenterol 40(11): 1321-1327; Sturniolo, G. C., et al. (2002). "Effect of zinc supplementation on intestinal permeability in experimental colitis." J Lab Clin Med 139(5): 311-315.
Answer: Thank you for your comments and valuable suggestions. I adopted the important references you mentioned to make the article more complete (line 172-174, page 4; line 261-269, page 6).
Q3. Lines 494-495: Please specify if the reported study is based on an animal study or human data. It would also be interesting to know at this point how much zinc was administered.
Answer: This study by Kirchhoff, P et al (ref 118) included isolation of gastric glands from both animal (rats) and human. Rats were given zinc salt (150 or 0.5 mg / kg/ day ZnCl) orally in their study. I added the dosage of zinc salt in the text (line 396-397, page 8)
Q4. Line 498-499: Is it 26.2 mg zinc or 26.2 mg zinc gluconate?
Answer: Thank you for your reminding. It is 26.2 mg zinc. To make this description clearer, I used containing 26.2 mg zinc instead of 26.2 mg zinc in the content (line 529, page 11).
Q5. Line 524: I suggest including a newer reference for the recommended daily intake of zinc, as this reference is already outdated. Both WHO and EFSA have changed their recommendations depending not only on the gender but also on the zinc bioavailability of the respective diet consumed (i.e. depending on the phytate content) (EFSA Panel on Dietetic Products, Nutrition and Allergies (NDA). Scientific opinion on dietary reference values for zinc. EFSA Journal 2014 2014, 12; World Health Organization / Food and Agricultural Organization. Vitamin and Mineral Requirements in Human Nutrition. 2 ed.; World Health Organization: Geneva, Switzerland, 2004).
Answer: Thank you for your comments. I delete the outdated reference (original reference 131), and added detailed description about the recommendation of daily zinc intake based on the updated information (both EFSA journal and WHO document (2nd edition) (line 553-564, new added references 136, 137) .
Q6. Lines 528-529: Please include a reference for the upper limit for zinc.
Answer: Thank you for your valuable suggestion. I added description about the issue of upper limit for zinc (line 562-564, page 11) (based on World Health Organization: Geneva, Switzerland, 2004).
Q7. Line 85: I believe mumol/L should be µmol/L.
Answer: Thank you for your correction. “µmol/L” was used instead of “.mumol/L” in the revised article (line 91, page 2)
Q8. Line 113: Please change systemic to systematic.
Answer: Thank you for your correction. “Systematic” was used instead of “systemic” in the revised article (line 118, page 3)

Reviewer 2 Report
This review article authored by Dr. Chao provides a comprehensive description of the current literature review and understanding of a topic with such enormous information on zinc supplementation in pediatric gastrointestinal illness.
I do suggest the following to be considered in improving this manuscript.
1. Shortening this manuscript to a readable level. The manuscript has made efforts to include all possible information in the literatures. Similar data or descriptions could be integrated together with a focused message (rather than to be very accurate and inclusive for every past publication). Clinicians as the target audience may not have patience or time to read all the information.
2. "3.4.1 Zinc-containing compounds and their gastroprotective effects section"--- This section could be better presented with a table listing the different compounds and key information (with advantages or disadvantages, and cost/availability in each use). A table will give readers more essential concise information to look it up if in need.
3. Section 4. (Line 520) Zinc supplements... "Authors should discuss the results and how they can be interpreted from the perspective of previous studies and of the working hypotheses. The findings and their implications should be discussed in the broadest context possible. Future research directions may also be highlighted." I am not sure these sentences are relevant in the beginning of this paragraph.
4. 6. Conclusion section (Line 560). This section could be more concise and to the points with take-home messages. It may be better described with a different language or description from the abstract section.
Author Response
Reviewer 2
This review article authored by Dr. Chao provides a comprehensive description of the current literature review and understanding of a topic with such enormous information on zinc supplementation in pediatric gastrointestinal illness.
Answer: Thank you for your positive feedback and comments.
I do suggest the following to be considered in improving this manuscript.
Q1. Shortening this manuscript to a readable level. The manuscript has made efforts to include all possible information in the literatures. Similar data or descriptions could be integrated together with a focused message (rather than to be very accurate and inclusive for every past publication). Clinicians as the target audience may not have patience or time to read all the information.
Answer: Thank you for your suggestion. I was recommended to add new information (including severalreferences) in the text bead on the other reviewer’s opinion. I have revised and rephrased many sentences to present my ideas clearer, and tried to shorten the article as possible.
Q2. "3.4.1 Zinc-containing compounds and their gastroprotective effects section"--- This section could be better presented with a table listing the different compounds and key information (with advantages or disadvantages, and cost/availability in each use). A table will give readers more essential concise information to look it up if in need.
Answer: Thank you for your valuable suggestions. I added a table to summarize the clinical utility of various zinc compounds (Table 1) in the revised article (line 405-406 & table 1 in the end of the text).
Q3. Section 4. (Line 520) Zinc supplements... "Authors should discuss the results and how they can be interpreted from the perspective of previous studies and of the working hypotheses. The findings and their implications should be discussed in the broadest context possible. Future research directions may also be highlighted." I am not sure these sentences are relevant in the beginning of this paragraph.
Answer: Thank you for your kind reminder. Indeed, this narrative is redundant, I deleted this paragraph.
Q4. 6. Conclusion section (Line 560). This section could be more concise and to the points with take-home messages. It may be better described with a different language or description from the abstract section.
Answer: Thank you for your comments. I revised the conclusion section to make this section more concise and to the point (line 613-618, page 13).
